# Effects of exchange vs. controlled diet on biochemical, body composition and functional parameters in elite female soccer players

**Sandra Antón San Atanasio** \*, **Sergio Maroto-Izquierdo** , **Silvia Sedano**

Department of Health Sciences, European University Miguel de Cervantes, Valladolid, Spain

\* santon@uemc.es

## Abstract

Due to the increasing level of professionalism, the high frequency of competitions, and the alarming injury rate observed in elite female soccer players, multidisciplinary strategies, including nutritional monitoring, need to be implemented. This study aimed to quantify energy, macronutrient and micronutrient intakes during the competitive period and to analyze the effects of two different nutritional interventions on nutritional knowledge, anthropometric data, biochemical values and physical performance. 19 elite female soccer players were randomly divided into two groups: the *controlled-diet group* (CG, n = 10), that followed a diet based on pre-established menus, and the *exchange-diet group* (EG, n = 9), that designed their own menus with an exchanged list. A cross-sectional study was designed to evaluate the dietary intake, while an experimental randomized controlled trial was designed to compare the effects of both 12-week nutritional interventions. Total energy, CHO, PROT, fibre and micronutrients intakes were below the general recommendations for athletes while, total and saturated fat intakes were above these. Moreover, there were no differences in diet during weekdays, pre-competition and competition days. The study also revealed a low nutritional knowledge and exchanged diet has demonstrated to be a better strategy to improve this. Biochemical monitoring showed that participants presented decreased concentration of haemoglobin and controlled diet may lead to greater effects on haemoglobin concentration and in anemia prevention. Both EG and CG showed significant reduction on skinfolds sum after intervention, but no significant differences were observed in thigh and calf indices. However, no significant changes were observed in soccer-related skills for any group.

## Introduction

Soccer is likely the most popular team sport, with over 265 million FIFA-registered players worldwide [1]. According to the UEFA, total number of female players in Europe has augmented by 220% in the last five years [2]. This high participation, together with its challenging physical demands, characterized by intermittent short bouts of high-intensity activities alternating with longer periods of low-to-moderate intensity aerobic exercise [3], has led to a considerable increase of scientific interest lately. In addition, due to the increasing level of

**Data Availability Statement:** All relevant data are within the paper and its attached files (figures).

**Funding:** The authors received no specific funding for this work.

**Competing interests:** The authors have declared that no competing interests exist.

professionalism, the high injury rate [4] and the high frequency of competitions [2], female players require the implementation of multidisciplinary strategies to optimize physical performance and health. Nutritional assessment and monitoring need to be consider and implemented [5].

Sports nutrition aims to satisfy energy and nutrient demands while optimizing physiological adaptations and guaranteeing recovery [6]. Nutritional requirements for high-intensity-intermittent exercise should include appropriate carbohydrate (CHO), protein (PROT), and total energy intake [5, 7]. Regarding CHO, recommendations for athletes range between 5–7 gCHO·kgBM$^{-1}$·day$^{-1}$ and 12 gCHO·kgBM$^{-1}$·day$^{-1}$ for intense competitive periods [8]. Whilst PROT requirements range between 1.2 and 2.5 gPROT·kgBM$^{-1}$·day$^{-1}$, depending on individual factors, such as training goal, age and health status. Moreover, total fat intake should be 20–35% of the total energy intake, where saturated fat should not exceed 10% of the total energy intake [6, 9, 10]. Particularly in soccer, female players have shown lower CHO intake [11–14] compared to their requirements. However, recorded PROT intake appears to be adequate (1.7–2 gPROT·kgBM$^{-1}$·day$^{-1}$) [12–14] which is even above the minimum requirements established. Previous studies demonstrated optimal values for fat intake, although the recommendation for saturated fat was generally exceeded [11–14]. Regarding micronutrient intake, recommendations for athletes are in line with those for the general population [6, 7]. However, female soccer players have previously registered deficiencies in iron, vitamin D, magnesium, folate, calcium, and vitamin E [12, 13]. It is worth mentioning that among the most representative functions of these micronutrients, the maintenance of hematopoietic functions, bone health, muscle function and the antioxidant capacity stand out [13].

In general, body composition, endocrine environment, immune response, and cognitive function may be influenced by energy intake management [6]. Hence, nutritional education is a common strategy used to improve athletes' dietary intake [9]. However, previous research suggested that soccer players have poor nutritional knowledge (NK) thus far [9]. Moreover, nutritional intervention studies in athletes, specifically in women's soccer, are limited, and dietary interventions have been described without detail. Bangsbo et al. [10] stated that to improve the players' nutritional habits, they should be aware of the contents of their diet. Thus, controlling the menus offered to them without nutritional education does not appear to be an effective method for optimizing their dietary intake [15]. In this line, Abood et al., [11] demonstrated that a nutritional education program for female college athletes, including soccer players, enhanced NK and changed dietary intake. However, the methodology is not clearly explained and is difficult to replicate.

There are different methods and types of nutritional intervention strategies that can be used with populations. For instance, controlled diets indicate what to eat each day or provide menu and food mix options for different intakes without the possibility of making changes. On the other hand, exchange diet systems offer different alternatives based on food exchange lists that approximately contribute the same macronutrient value and energy. They can be exchanged in meal planning without significant differences in dietary intakes, providing subjects greater flexibility [16].

To promote adherence to optimal nutritional habits in female soccer players, the design and implementation of a nutritional intervention requires *a priori* knowledge of dietary intake to identify specific requirements and to develop a dietary strategy with a given methodology. To our knowledge, no well-designed nutritional education program has been implemented in female soccer players. Therefore, this study aimed to a) quantify the energy, macronutrient, and micronutrient intakes of female soccer players during the competitive period and compare them to general recommendations, and b) analyze the effects of two different nutritional interventions (i.e., exchange diet vs. controlled diet) on NK and adherence, anthropometric data,

biochemical values, and physical performance. We hypothesized that the exchange diet may be the better approach to improve female soccer players' NK while generating higher adherence to the dietetic plan. However, no differences between the exchange diet and controlled diet are expected to be observed on anthropometric, biochemical, and physical performance outcomes.

## Methods

### Participants

The study included 19 female elite soccer players who voluntarily participated and were divided into two groups: the "controlled-diet group" (CG) (n = 10; 19 ± 1.4 years; 63.6 ± 10 kg; 164 ± 2.0 cm) and the "exchange-diet group" (EG) (n = 9; 22.5 ± 4.8 years; 61.3 ± 7 kg; 164 ± 0.1 cm). A simple randomization was performed by generating a table of random numbers without repetition. Even numbers were assigned to the CG and odd numbers to the EG. All participants belonged to the same Spanish team and performed the same training program, which included 8 hours per week of practice and one official match per week. During the intervention players were not allowed to engage in any other training, to follow a different nutritional plan or to use any ergogenic aid that could affect the results. Prior to the start of the intervention, the players were fully informed about the purposes and risks of the study and provided written informed consent. They also completed a form with their personal, medical, and training details. The research project was conducted in accordance with the Declaration of Helsinki and was approved by the University Review Board for use of Human Subjects (Code: UEMC 10_2021) (Fig 1).

### Design and procedures

A cross-sectional study was designed to evaluate the dietary intake and NK during the competitive season in elite female soccer players. On the other hand, an experimental randomized controlled trial was designed to compare the effects of two different 12-week nutrition education interventions on NK, adherence, anthropometric data, physical performance, and biochemical indices. The intervention was performed at the start of the second part of the competitive season, and all the tests were conducted at two times points: pre-intervention (PRE) (i.e., one week before the start of the intervention) and post-intervention (POST) (i.e., one week after the end). Dietary intake was only recorded at PRE.

After baseline assessments, participants were informed about their specific nutritional plan. The main researcher conducted the educational session (60 minutes) in a relaxed atmosphere using a slide keynote focused on the importance of nutrition for soccer performance and explaining the specific intervention on each group. Both group diets were designed to provide the same amount of energy (Kcal) and nutrients (PROT, CHO and fat) at a similar timing. Total energy needs were calculated taking into account the average weight of the players in the study (60 kg) (42 ± 3 Kcal·kgBM$^{-1}$·day$^{-1}$) [17]. Nutrient needs were estimated for each player based on recent recommendations for team sports [7] and adjusted according to their individual weight and actual dietary intake recorded in the diaries. The recommended daily intake for PROT, CHO and fat were 1.5 g·kgBM$^{-1}$·day$^{-1}$, 4 g·kgBM$^{-1}$·day$^{-1}$ and 0.7 g·kgBM$^{-1}$·day$^{-1}$, respectively. For instance, CHO intake in the intervention was slightly lower than the minimum current recommendation range of 5–7 gCHO·kgBM$^{-1}$·day$^{-1}$ [6, 8, 18] due to the participants' baseline CHO intakes being less than 3 gCHO·kgBM$^{-1}$·day$^{-1}$. A sudden increase in CHO intake would have been difficult to adhere to. PROT intake timing was based on 0.3 g·kgBM$^{-1}$·day$^{-1}$ post-exercise and every 3–5 hours [6]. Table 1 displayed the distribution and the timing.

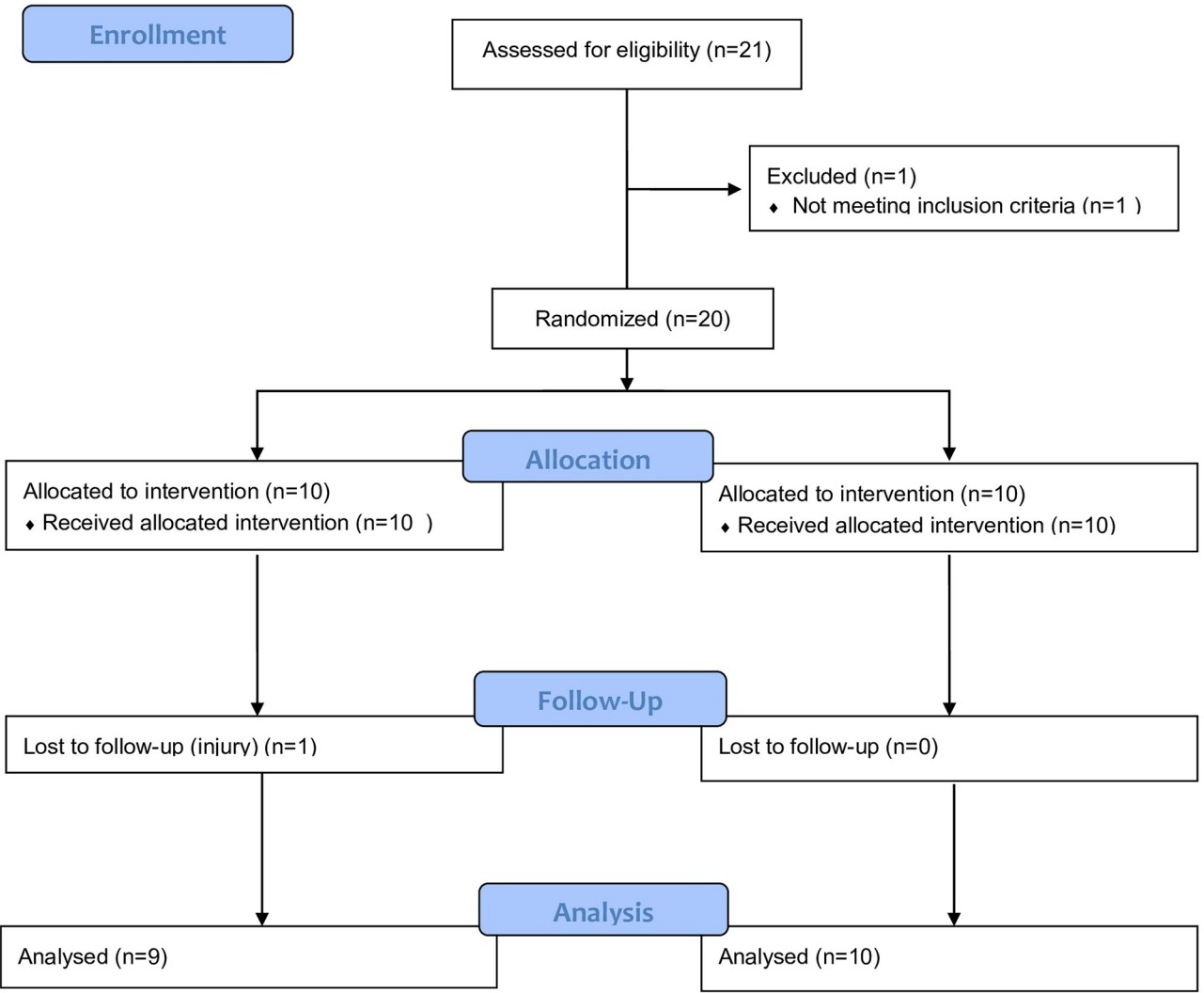

**Fig 1. Participants flow diagram.**

**Table 1. Energy and nutritional diet distribution during the intervention (CG and EG).**

|  | CHO (g) | PROT (g) | Fat (g) | Energy (Kcal) |
|---|---|---|---|---|
| Breakfast | 50 | 15 | 7 | 323 |
| Lunch | 40 | 15 | 7 | 283 |
| Main meal* | 70 | 22.5 | 10 | 460 |
| Snack | 40 | 15 | 7 | 283 |
| Dinner | 50 | 22.5 | 10 | 380 |
| TOTAL | 250 | 90 | 41 | 1729 |

*(lunch in this Mediterranean population)

During the intervention, CG followed a diet consisting of pre-established menus and food combinations. Each day, players had to select one option from four different breakfasts, seven snacks/lunches, seven main meals and seven dinners. They were not allowed to substitute any food or ingredients. They were encouraged to vary their choices from the pre-established menu options. Different options were equivalent among them. In contrast, EG players were given exchange lists and instructed to create their own menus within the specified guidelines. The exchange list included food equivalents that could be swapped out for each other. The exchange list included CHO and PROT equivalents which provided a consistent amount of nutrients per serving: the exchangeable CHO equivalents provided 10 g of CHO, while the exchangeable PROT equivalents provided 7 g of PROT. EG players were able to make daily adjustments to their menus and exchange food and ingredients as needed. Fats and oils used in the cooking and dressings were not included in the dietetic indications because the general recommendations were the same in both groups. During the entire intervention, all subjects had access to a professional nutritionist to address any related questions.

## Measures

**Dietary intake, nutritional knowledge and adherence questionnaires.** Dietary intake was measured using a 7-day food record sheet that included the day of competition (Sunday), and the day before (Pre-competition, Saturday) and five weekdays (Monday to Friday) [19]. For the weekdays, the mean intake across the five days was used for analysis. Players were previously educated about the protocols for documenting intake and a researcher assisted them with the accuracy of their self-reported information. These intakes were, then, averaged and represented as the pre-intervention average intake. Total energy intake, PROT, CHO, fat, fibre, vitamins and minerals were analyzed using Evalfinut App (Ibero-American Foundation, 2015) following the Spanish Food Composition Database (BEDCA) [20] and European dietary reference values for nutrients (EFSA) [21].

NK was assessed using the General Nutrition Knowledge Questionnaire (NKQ) [22]. It consisted of 32 questions that were reviewed, translated and adapted. A score of 1 was given for each correct answer whereas a score of 0 was given for each incorrect or "unsure" response, allowing for a total possible score of 32 points. The individual result was reported as percentage. Overall performance in the NKQ was established in five categories, based on the scoring systems used by other authors [9]: very poor knowledge ($< 25\%$), poor knowledge (25–49%), average knowledge (50–65%); over average knowledge (66–74%); excellent knowledge (75–100%). On the other hand, a diet adherence questionnaire (AQ) was created *ad-hoc* for this study and it is provided as an attachment. It consisted of four questions formulated according to the diet strategy used on each group. Similarly to the Likert scale [23], the scoring system used in this study to assess diet compliance and the tracking difficulty was of 0–4 and 0–10, respectively. It also included questions about difficulties achieving general and specific nutrient intakes. Both questionnaires (NKQ and AQ) were completed anonymously and individually at their own places.

**Anthropometric measurements.** All the anthropometric measurements were taken in accordance with the standardized procedures of the International Society for the Advancement in Kinanthropometry [24]. Height (cm) and body mass (kg) were measured for each player using a SECA® stadiometer (240 –model) and a TANITA® balance (BF– 666, Germany). 6 skinfolds (triceps, subscapular, suprailiac, abdominal, front thigh, and medial calf) were measured with a Holtain limiting calliper (British Indicators® Ltd) and the perimeter of the thigh and calf were measured with an inextensible tape (SECA®). Subjects were tested at

the same time of the day in the PRE and POST. All anthropometric measures were highly reliable with ICC of 0.94–0.97 (95% CI) for skinfolds and 0.93–0.98 (95%) for perimeters.

**Physical performance.**    Running speed and vertical jump performance were assessed 2 days before and after the intervention to analyze the diet-induced effect. After a protocolled warm-up, participants were requested to perform 3 repetitions of a 20-m maximum sprint, starting from a total static position 1 m behind the start line on the competitive surface (i.e., artificial turf) with the competitive equipment (i.e., soccer cleats) using two photocell gates (Racetime2, Microgate, Italy) [25]. A 1-min recovery period was allowed between repetitions. After a recovery period of 3 minutes, vertical jump performance was measured on a contact mat (Globus Ergo Tester, Codogne, Italy) using the countermovement jump with arms swing (CMJA) [26]. Participants were requested to perform 3 maximal attempts with 1 minute of recovery between repetitions. The best result of each test was selected for further analysis. The ICC was 0.95 (95% CI) for CMJA and 0.97 (96% CI) for 20-m sprint.

**Biochemical analysis.**    Blood analyses were performed 2 days before and after intervention at the same time of the day in a fasted and rested state. Data were obtained using an electronic hematology analyzer (Sysmex–SE9500, TOA Medical Electronics, Kobe, Japan). The outcomes measured were haemoglobin and hematocrit level, urea, iron, transferrin, transferrin saturation index, ferritin, vitamin B12, vitamin D25 –OH and cortisol. The evaluation was based on the reference range (Table 3) indicated by the biochemical analysis laboratory.

**Statistical analysis.**    Statistical analyses were performed using SPSS v.26.0 (SPSSInc. Chicago, IL). Results are expressed as mean ± SD. Data distribution was examined for normality using the Shapiro–Wilk test and results revealed that all the variables presented a normal distribution of the data. A mixed-measures analysis of variance (ANOVA) with one between-subjects factor (intervention condition: EG or CG) and one within-subjects factor (time) followed by Bonferroni post-hoc tests was used to investigate differences in variables measured. The effect size (ES) was calculated for interactions between groups using Cohen's guidelines. Threshold values for ES were > 0.2 (small), > 0.6 (large) and > 1.2 (very large) [27]. The significance level was set to $p < .05$. In addition, the reliability of measurements was calculated using the ICC.

The sample size for the study was determined a priori using G*Power 3.1.9.2 (Heinrich Heine-Universität Dusseldorf, Dusseldorf, Germany; http://www.gpower.hhu.de/) for an ANOVA repeated measures design [28]. The effect size was calculated based on the means and between-subject standard deviations obtained from a previously published study [29] that examined the impact of a nutritional intervention on nutritional status, natural killer (NK) cells, body composition, and performance in athletes. The mean values for vertical jump performance were 71.0 ± 6.1 cm and 75.7 ± 7.1 cm, respectively. The resulting mean difference and average standard deviation were 5.1 cm and 5.1 cm, yielding a Cohen's d_z effect size of 0.79, which can be classified as moderate (equivalent to a moderate effect size, f = 0.3). The average standard deviation was used to calculate the effect size [30]. The significance level (alpha) was set at 5%, and the desired power (1-β) was set at 80% [31]. The estimated sample size was 17 participants, resulting in an actual power of 0.802. To account for potential dropouts, we enrolled 21 participants in the study. Importantly, it should be noted that the sample was highly representative of the study population, as it included 19 players (2 players finally dropped out) out of a total of 327 registered female soccer players in the highest category of Spanish soccer [32].

## Results

Nutritional intake is showed in Table 2. The repeated ANOVA revealed no significant differences in any nutrient among 5 weekdays, pre-competition and competition days. Total energy

Table 2. Nutritional intake of Spanish elite soccer players (mean ± SD) and current recommendations [6, 8, 18].

| Nutrient intake | 5 weekdays *** (from Monday to Friday) | Pre-competition (Saturday) | Competition (Sunday) | Recommendation ** |
|---|---|---|---|---|
| Energy (Kcal) | 1590 ± 539 | 1710 ± 735 | 1657± 487 | 2520–2880[1] |
| Carbohydrates (g) | 180 ± 86 | 177 ± 86 | 167 ± 71 | NA |
| Carbohydrates (g·kgBM$^{-1}$·day$^{-1}$) | 2.89 ± 1.38 | 2.83 ± 1.37 | 2.68 ± 1.14 | 5–7 (g·kgBM$^{-1}$·day$^{-1}$) |
| Total fat (g) | 62 ± 25 | 73 ± 34 | 78 ± 24 | NA |
| Total fat (% of E)* | 35.21 | 38.59 | 42.34 | 20–35% of E |
| Saturated fat(g) | 23 ± 10 | 28 ± 11 | 21 ± 8 | NA |
| Saturated fat (% of E) | 12.89 | 14.58 | 11.51 | < 10% of E |
| Mono-unsaturated fat (g) | 22.26 ± 8.73 | 25.80 ± 13.96 | 30.09 ± 12.08 | NA |
| Poly-unsaturated fat (g) | 9.77 ± 6.14 | 12.04 ± 7.58 | 14.65 ± 6.63 | NA |
| PROT (g) | 70 ± 26 | 71 ± 25 | 69 ± 18 | NA |
| PROT (g·kgBM$^{-1}$·day$^{-1}$) | 1.12 ± 0.42 | 1.14 ± 0.41 | 1.11 ± 0.28 | 1.2–2.5 (g·kgBM$^{-1}$·day$^{-1}$) |
| Fiber (g) | 15 ± 13 | 10 ± 5 | 10 ± 4 | 22–25 g·day$^{-1}$ |
| Calcium (mg) | 629 ± 258 | 594 ± 302 | 449 ± 175 | 800–1000 |
| Phosphorous (mg) | 1083 ± 380 | 928 ± 293 | 717 ± 206 | 700 |
| Magnesium (mg) | 199 ± 92 | 168 ± 55 | 156 ± 51 | 330–300 |
| Iron (mg) | 11 ± 5 | 10 ± 5 | 9 ± 3 | 18 |
| Vitamin C (mg) | 59 ± 47 | 42 ± 39 | 32 ± 31 | 60 |
| Vitamin E (mg α-tocopherol) | 6 ± 4 | 5 ± 4 | 7 ± 3 | 12–15 |
| Vitamin D (μg) | 3 ± 3 | 2 ± 2 | 2 ± 2 | 5 |
| Vitamin B12 (g) | 8 ± 16 | 5 ± 5 | 13± 6 | 2 |

*% of E: percentage of total energy intake

**Recommendation: current recommendations for daily average intake. NA: not applicable. (Tomas et al., 2016; Ranchordas et al., 2017; FESNAD, 2015)

*** For the weekdays, the mean intake across the five days was used for analysis

[1] Calculated taking into account the average weight of the players in the study (60 kg) (42 ± 3 Kcal·kgBM$^{-1}$·day$^{-1}$) (Gibson et al., 2011).

intake, CHO, PROT and fibre were shown to be below the current recommendations. On the contrary, total fat and saturated fat intake were above them. Regarding micronutrients, vitamin C, vitamin E and vitamin D, calcium, magnesium and iron intakes were also below recommendations. Only phosphorous and vitamin B12 intakes were considered as adequate.

NK was classified as "average" knowledge (60–61%) for both EG and CG before intervention. However, the EG showed significant increases in NK ($p < 0.05$, 7%; NK classified as "over average") after intervention. No significant differences between groups were observed.

Diet adherence questionnaire (AQ) showed significant group x time interaction in the CG, showing a higher compliance. However, a significant decrease in adherence was observed in the EG after intervention. POST-Intervention general difficulty was 6.5 in the EG and 5.9 in the CG. In general, throughout the intervention both groups demonstrated greater difficulties with PROT intake. In addition, breakfast was the most difficult meal to follow in CG during the intervention.

Group x time effects after both EG and CG interventions on physical and anthropometrical parameters were reported in Fig 2. Regarding anthropometrical data, both conditions showed significant skinfolds sum decreases after the intervention (Fig 2B; EG: $p < 0.001$; F = 25.0; ES = 0.58, - 15.3%; CG: $p < 0.01$; F = 15.3; ES = 0.49, -13.0%). However, no significant interactions were dissipated after the dietary intervention for EG ($p = 0.132$; ES = 0.31; 1.9%) nor CG ($p = 0.535$; ES = 0.03; 0.5%) when body mass was analysed (Fig 2A). In addition, no significant changes were shown for any group on the perimeters of the thigh (EG: PRE = 45.0 ± 2.2,

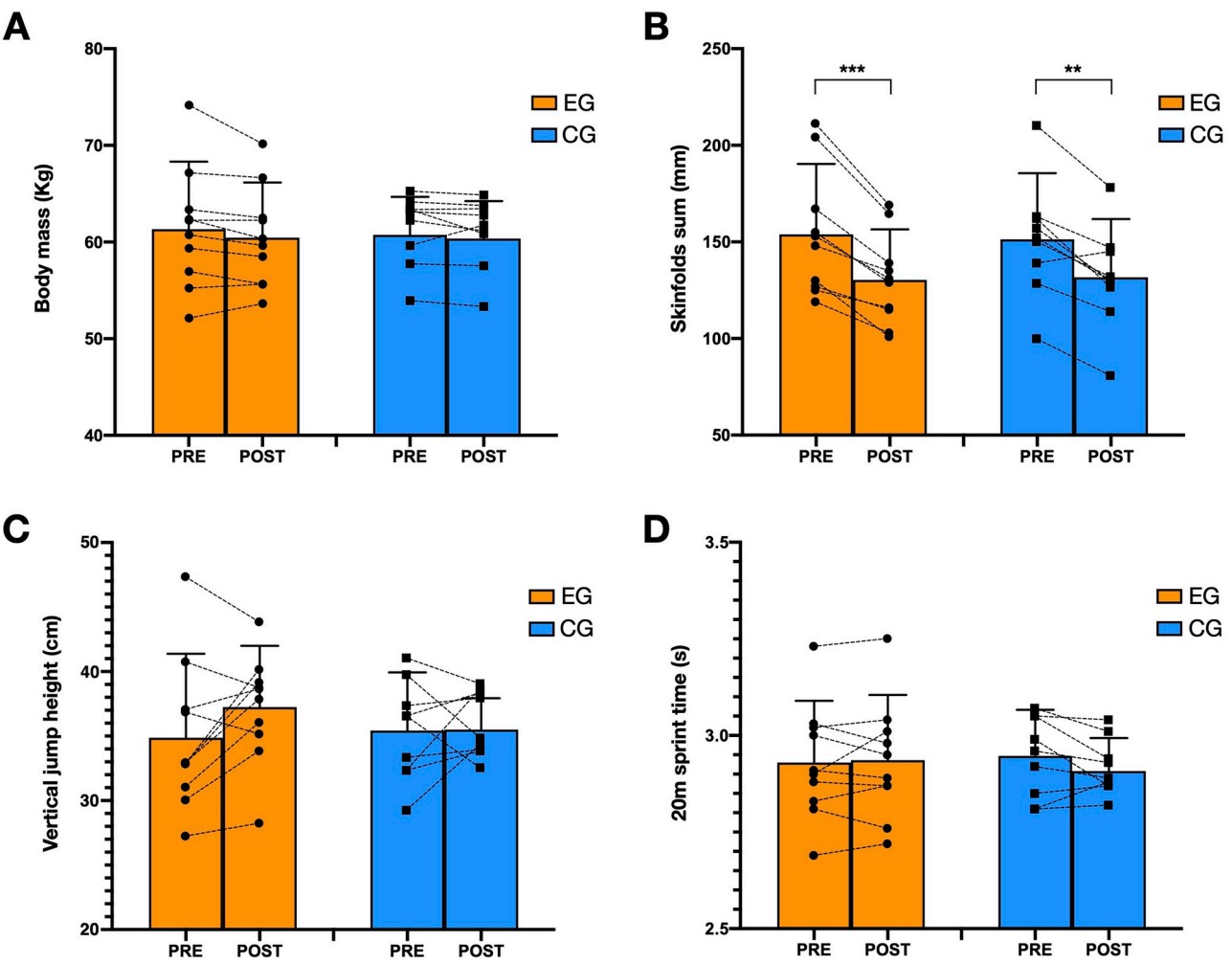

**Fig 2. Effects after both EG and CG interventions on anthropometrical and physical parameters.** (A) Body mass; (B) Skinfolds sum (mm); (C) Vertical jump height (cm); (D) 20 m sprint time (s). ***, Significant (p < 0.001); **, Significant (p < 0,01) differences between pre- and post—test measurements.

POST = 45.9 ± 2.9, p = 0.430; ES = 0.58, -15.3%; CG: PRE = 46.6 ± 5.2, POST = 46.9 ± 5.4, p = 0.654; ES = 0.49, -13.0%) and calf (EG: PRE = 36.0 ± 2.7, POST = 36.1 ± 2.5, p = 0.782; ES = 0.02, 0.2%; CG: PRE = 35.9 ± 3.0, POST = 36.1 ± 2.7, p = 0.556; ES = 0.04, 0.4%). The repeated ANOVA did not report meaningful differences between EG and CG interventions for CMJA height (Fig 2C; EG: p = 0.144; ES = 0.31; 6.5%; CG: p = 0.574; ES = 0.23, 2.8%) and on 20 m sprint time (Fig 2D; EG: p = 0.767; ES = 0.03; 0.2%; CG: p = 0.135; ES = 0.33, -1.4%).

Biochemical variables are displayed in Table 3. The repeated ANOVA revealed significant time x group interaction effects only on haemoglobin for the CG (p < 0.001; F = 20.3; ES = 1.07, 3.6%) and on cortisol for the EG (p < 0.05; F = 5.1; ES = 0.73, 36.4%). No significant differences were found between groups for any biochemical variable at any time.

## Discussion

Given the growing popularity of women's soccer, its physically demanding nature and the high incidence of injuries, various approaches to optimize performance and decrease injury

**Table 3. Changes (mean ± SD) in biochemical variables for the CG (n = 10) and EG group (n = 9) before (PRE) and after intervention (POST).** P value for the comparison between pre- and post-intervention values by Bonferroni test, and effect size (ES) for the changes, the magnitude of change (%), as well as the reference values are shown for each group.

| | Group | PRE | POST | P | ES | % | Reference range |
|---|---|---|---|---|---|---|---|
| Haemoglobin (g.dL$^{-1}$) | EG | 12.78 ± 1.03 | 13.23 ± 0.92 | 0.076 | 0.38 | 3.6 | 11.70–15.70 |
| | CG | 12.57 ± 0.86 | 13.71 ± 0.89 | 0.001 * | 1.07 | 9.1 | |
| Hematocrit (%) | EG | 38.86 ± 2.73 | 39.74 ± 2.85 | 0.756 | 0.26 | 2.3 | 35.50–45.50 |
| | CG | 38.34 ± 2.63 | 36.64 ± 11.03 | 0.575 | 0.21 | - 4.4 | |
| Urea (mg.dL$^{-1}$) | EG | 33.75 ± 7.59 | 29.50 ± 3.96 | 0.198 | 0.53 | - 12,6 | 17.00–49.00 |
| | CG | 34.00 ± 6.53 | 33.00 ± 6.78 | 0.770 | 0.12 | - 2,9 | |
| Fe (μg.dL$^{-1}$) | EG | 78.14 ± 29.34 | 63.86 ± 31.33 | 0.377 | 0.39 | - 18.3 | 50.00–170.00 |
| | CG | 93.57 ± 27.02 | 63.43 ± 22.15 | 0.077 | 0.97 | - 32.2 | |
| Transferrin (mg.dL$^{-1}$) | EG | 249.86 ± 23.55 | 254.00 ± 18.03 | 0.369 | 0.15 | 1.66 | 200.00–360.00 |
| | CG | 261.00 ± 33.31 | 263.14 ± 25.92 | 0.638 | 0.06 | 0.82 | |
| Transferrin saturation (%) | EG | 25.29 ± 9.45 | 20.00 ± 9.57 | 0.288 | 0.45 | - 20.9 | 20.00–55.00 |
| | CG | 28.71 ± 9.83 | 19.29 ± 7.83 | 0.071 | 0.84 | - 32.8 | |
| Ferritin (ng.dL$^{-1}$) | EG | 22.25 ± 9.27 | 22.03 ± 12.52 | 0.951 | 0.02 | - 1.0 | 100.00–120.00 |
| | CG | 31.14 ± 15.76 | 24.86 ± 11.44 | 0.124 | 0.35 | - 20.2 | |
| Vitamin B12 (pg.dL$^{-1}$) | EG | 403.57 ± 126.64 | 431.57 ± 135.59 | 0.341 | 0.18 | 6.9 | 206.00–678.00 |
| | CG | 590.71 ± 245.16 | 571.14 ± 195.41 | 0.501 | 0.07 | - 3.3 | |
| Vitamin D (ng.dL$^{-1}$) | EG | 17.86 ± 4.49 | 18.63 ± 6.03 | 0.722 | 0.12 | 4.3 | 30.00–100.00 |
| | CG | 17.25 ± 7.21 | 19.37 ± 11.59 | 0.362 | 0.19 | 12.3 | |
| Cortisol (μg.dL$^{-1}$) | EG | 16.10 ± 6.88 | 21.96 ± 5.74 | 0.043 * | 0.73 | 36.4 | 5.00–23.00 |
| | CG | 22.20 ± 2.97 | 21.86 ± 5.29 | 0.897 | 0.07 | - 1.5 | |

Abbreviatures: EG, Exchanged-diet group; CG, Controlled-diet group.

*, Significant (p < 0.05) differences between pre- and post- test measurements.

risk and severity have been explored in the scientific literature. Athletes need to consume enough energy to ensure an effective metabolic and neuromuscular response, and thus, develop sport-specific demands [6]. A diet that contains insufficient energy can result in several performance detriments, including loss of lean mass and bone tissue, as well as an increased risk of overtraining and injury. It may also contribute to endocrine disturbances [33]. Hence, this study aimed to quantify energy, macronutrients and micronutrients intake during the competitive period in elite female soccer players, as well as, to compare two different nutritional interventions on NK and diet adherence, anthropometric features, biochemical values and physical performance. The main findings revealed that the total energy, CHO, PROT, fibre and some micronutrients intakes were below the current recommendations. On the contrary, total fat and saturated fat intakes were higher than those generally recommended. Players did not present differences in diet during weekdays, pre—competition and competition days. In addition, exchange diet has demonstrated to be a better strategy to increase players' NK. However, both EG and CG showed significant reduction on skinfolds sum after intervention, although no significant differences were observed in other anthropometrical variables (i.e., body mass and perimeters of the thigh and calf). Regarding biochemical values, CG has shown to increase haemoglobin concentration but no other parameters. The EG, however, only showed significant changes in cortisol level. Finally, no significant differences were observed in any physical performance tests between the two groups after either the exchange diet or the controlled diet, as hypothesized.

Regarding energy intake, several studies have also reported an insufficient intake in female soccer players [11–14], even below the requirement of 45–50 Kcal·kgBM$^{-1}$·day$^{-1}$ [34]. However, our study revealed that our players did not reach the minimum of 30 Kcal·kgBM$^{-1}$·day$^{-1}$, and this deficiency persists in pre-competition and competition days. This situation could cause physiological disturbances that impair players' health and performance. Regarding macronutrients intake, CHO intake is considered a key factor for performance in soccer [35]. Previous researches on female soccer players have shown that it reaches 55–60% from the total energy intake [11–14], which seems to be adequate. However, when normalized by body weight, CHO intake usually does not reach the minimum consumption required for optimal performance. For instance, in this study, the intake was ~ 45% of total energy intake, bellow 3 gCHO·kgBM$^{-1}$·day$^{-1}$, far from the recommended 5–7 gCHO·kgBM$^{-1}$·day$^{-1}$ [8]. PROT intake was also below the general recommendations (1.2 gPROT·kgBM$^{-1}$·day$^{-1}$), supposing a higher risk of deficiency not only regarding the optimal energy intake, but also limiting the ability to recover after practice/competition and the potential anabolic state [36]. Nevertheless, similar studies revealed that the PROT intake in female soccer players is usually adequate with a trend to increase in recent years, reaching 1.7–2 gPROT·kgBM$^{-1}$·day$^{-1}$ [11–14]. On the contrary, total fat intake was above the general recommendations in our sample, while other studies revealed proper fat consumption in female soccer players [11–14]. Similarly, previous investigations have also observed an excess in the saturated fat intake [12–14]. Female soccer players must first meet the demands of CHO and PROT, while fats must supplement the energy value of the daily intake [37]. Finally, fibre intake is also below the general recommendations, as previously shown by others [13, 14]. On the other hand, iron, calcium, vitamin D and antioxidant deficiencies are more prevalent in women than men [6]. In this line, diet evaluation and biochemical monitoring may facilitate diagnosis of micronutrients deficiencies. In our study, participants presented deficiency on haemoglobin, ferritin and vitamin D, which could be related to the insufficient intake in some micronutrients. Although these deficiencies are usual in female soccer players [12, 13], they may imply not only a decrease in performance and an increased risk of injury, but also affect the general health of athletes (e.g., anemia). Therefore, it seems to be necessary to implement an individualized supplementation prescription combined with dietary instructions to improve micronutrients assimilation.

When implementing any nutritional intervention, previous individual NK should be considered [27–30]. In this study, players' NK was classified as "average" for both groups before intervention. However, it significantly improved after intervention in EG. Although studies analyzing NK in women's soccer are scarce and the strategies employed in some of them are not clearly explained, exchange diet has been proposed to be an optimal strategy to improve NK in female soccer players [11]. This is in line with our results, confirming that any given nutritional intervention is more efficient when linked to a nutritional education program [15]. Generally, it has been shown to increase participant's adherence to diet [38], although it was not observed in this study. However, on the light of our results, it was evidenced that female soccer players showed the greatest difficulty to reach the recommendation for PROT intake. This was likely related to the daily inadequate PROT distribution showed at pre-intervention.

It is widely known that nutritional interventions lead to significant anthropometric modifications, which are directly related to athletes' performance. In the case of our intervention, both EG and CG were intended for modify fat mass, since it exceeded previous registers of professional and semi-professional female soccer players [39]. However, results revealed that both EG and CG led to similar results in anthropometric features, decreasing the total skinfold sum. This may be due to the fact that both EG and CG interventions were energetically and nutritionally similar, and all the players followed the same training plan. However, no significant differences were observed in body mass and thigh and calf perimeters. This could be

explained because players could have maintained or even increased their lower limbs lean muscular mass.

Vertical jump and sprint performance have been shown to be directly related to body composition [40]. Rossie et al. [29] observed a significant correlation between changes in sport-related skills (e.g., sprinting) and the percentage of body fat after a nutritional intervention, showing better sprint results in those players with lower fat percentage. However, no significant changes were found in CMJ or 20 m sprint time after intervention in our study for any group. This relationship between sport-specific skills performance and body composition in female is more difficult to observe due to the endocrine fluctuation that promotes the menstrual cycle.

In addition to the abovementioned large-scale changes induced by the nutritional approach, large changes also occur at biological level. According to our results in biochemical variables, significant differences were only found in cortisol and haemoglobin. Cortisol is considered a marker of exercise-related stress and balance between anabolic and catabolic processes. The promoted stress at the end of the season, the highest competitive demand and decisive competitions may impact on cortisol levels [41]. Thus, the results of this study regarding cortisol levels have been attributed to those coadjutant situations that may affect its concentration. On the other hand, it is known that haemoglobin values can be improved by using a controlled nutritional strategy that takes into account the optimization of iron absorption through the incorporation of foods rich in vitamin C, which promotes optimal biochemical values related to the production of red blood cells [42]. Although the intake of iron and vitamin C was not monitored, this could explain the difference found, since EG did not take this optimization into account. Based on this, it could be beneficial to accompany exchange diets with recommendations related to this aspect, particularly in populations at risk of anemia.

Several potential delimitations are worthy of mention. The study was conducted over 12 weeks, and whilst relatively some changes in anthropometric variables and some biochemical values were observed, it would be of interest to test long-term effects after longer intervention periods and in other outcomes, such as measurement of body composition by a gold standard method (e.g., Dual–energy X ray absorptiometry) and evaluation of post-intervention dietary intake. In addition, although the sample was homogeneous with respect to age, sex, training level and training experience, it remains unknown whether similar results would be found in female soccer players of different chronological or training ages. Therefore, future studies need to improve methodological quality (e.g. assessment of adherence) and limit confounders (eg. menstrual cycle) to facilitate a deeper understanding of the effects of a nutritional intervention.

The present study revealed the low nutritional knowledge shown by professional soccer players, which may affect their performance and health, since their total energy, CHO, PROT, fiber, and micronutrient intakes were below the current recommendations, while their total fat and saturated fat intakes were higher than those generally recommended. Exchanged diet was found to be more effective in increasing the nutritional knowledge of players. On the other hand, a 12-week controlled diet intervention was associated with an increase in haemoglobin concentration in female soccer players, which may play a crucial role in preventing anemia. Both dietary interventions resulted in a significant decrease in the sum of skinfolds, however, their effects on lean mass are still unknown.

As previously stated, the participants' baseline carbohydrate intake was less than the recommended amount. Considering that female athletes have a high incidence of eating disorders and a strong focus on weight control [6, 7], the carbohydrate intake during the intervention was slightly lower than the current minimum recommendation. This was done to avoid a sudden increase in intake that could be difficult for the participants to adhere to. Gradually

increasing carbohydrate intake can help the body adjust and make dietary changes more sustainable in the long term. However, this could be considered a limitation of the study, and future research should take into account the actual needs of participants, with their intake during the intervention aligning with current scientific-based recommendations. Additionally, further research is needed to analyze the effects of both exchanged and controlled diets on athletes in different phases of the competitive season, as well as in athletes with a higher level of nutritional knowledge at baseline.

## Supporting information

**S1 Questionnaire. This is the adherence questionnaires.**
(DOCX)

**S2 Questionnaire. This is the translation of adherence questionnaires.**
(DOCX)

**S1 Database. This is the database of Fig 2.**
(XLSX)

**S2 Database. This is the global database.**
(XLSX)

## Author Contributions

**Data curation:** Sandra Antón San Atanasio, Sergio Maroto-Izquierdo, Silvia Sedano.

**Formal analysis:** Sergio Maroto-Izquierdo, Silvia Sedano.

**Investigation:** Sandra Antón San Atanasio, Sergio Maroto-Izquierdo, Silvia Sedano.

**Methodology:** Sandra Antón San Atanasio, Sergio Maroto-Izquierdo, Silvia Sedano.

**Project administration:** Sandra Antón San Atanasio.

**Resources:** Sandra Antón San Atanasio.

**Validation:** Sergio Maroto-Izquierdo, Silvia Sedano.

**Visualization:** Sandra Antón San Atanasio, Sergio Maroto-Izquierdo, Silvia Sedano.

**Writing – original draft:** Sandra Antón San Atanasio, Sergio Maroto-Izquierdo, Silvia Sedano.

**Writing – review & editing:** Sandra Antón San Atanasio, Sergio Maroto-Izquierdo, Silvia Sedano.

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
