## [Editor Report · Decision Letter 0]

29 Sep 2022

PONE-D-22-26202Effects of exchange vs. controlled diet on biochemical, body composition and functional parameters in elite female soccer playersPLOS ONE

Dear Dr. Antón,

Thank you for submitting your manuscript to PLOS ONE. The initial review of your submission showed elements that should be improved / completed before considering proceeding your work for further peer review stagesTherefore, we invite you to submit a revised version of the manuscript.

ACADEMIC EDITOR:

Accurate calculations and their screens in terms of the correctness of the determination of the minimum sample size should be provided. The indicated sample size may refer to the size within one given group. In this case, numbers of 9 or 10 in the group would not be sufficient.

Accuracy of data should be reported in accordance with the accuracy of the method.

Thus, the energy supply in kcal should not have any decimal places at all. Similarly, for 3-digit nutrient intake values are overestimated - this should be corrected.

Correct the correct entries and editing - e.g. superscripts in some units (g·kg^-^^1^), data spacing etc.

Including a flowchart of the study design and procedure will be beneficial to consider.

We look forward to receiving your revised manuscript.

Kind regards,

Krzysztof Durkalec-Michalski, Ph.D

Academic Editor

PLOS ONE

"Thank you for stating the following financial disclosure: 

"no" 

"no"

---

## [Author Response · Author response to Decision Letter 0]

4 Nov 2022

- Manuscript PONE-D-22-26202: Effects of exchange vs. controlled diet on biochemical, body composition and functional parameters in elite female soccer players

Response to reviewers (ACADEMIC EDITOR). Please find our anwers, point by point:

Point 1 (Sample size) 

Accurate calculations and their screens in terms of the correctness of the determination of the minimum sample size should be provided. The indicated sample size may refer to the size within one given group. In this case, numbers of 9 or 10 in the group would not be sufficient.

Response: We appreciate this comment. We have made the pertinent modifications to clarify this point in the second paragraph of the Statistical Analysis section. In addition, it should be noted that the sample is highly representative of the study population, as it includes 19 players of a total of 327 federated female soccer players within the highest Spanish soccer level.

Point 2 (Accuracy of data)

Accuracy of data should be reported in accordance with the accuracy of the method.

Response: We have added the intraclass correlation coefficient (ICC) values in order to show the accuracy of the data. It can be found in the METHODS Section, both for anthropometric measurements and physical performance. Moreover, in the Statistical analysis section we have indicated that “the reliability of measurements was calculated using the ICC”.

Point 3 (decimal places)

Thus, the energy supply in kcal should not have any decimal places at all. Similarly, for 3-digit nutrient intake values are overestimated - this should be corrected.

Response: According to you comment, we have checked and corrected all the manuscript and tables.

 Point 4 (Correct entries and editing)

Correct the correct entries and editing - e.g. superscripts in some units (g·kg-1), data spacing etc.

Response: According to you comment, we have checked and corrected all the manuscript and tables.

Point 5 (Flowchart)

Including a flowchart of the study design and procedure will be beneficial to consider.

Response: As you suggested, we have added a flowchart of the study in order to clarify the information given. It has been included as Figure 1 in the METHODS section.

---

## [Decision Letter · Decision Letter 1]

10 Feb 2023

PONE-D-22-26202R1Effects of exchange vs. controlled diet on biochemical, body composition and functional parameters in elite female soccer playersPLOS ONE

Dear Prof. Antón,

Thank you for submitting your manuscript to PLOS ONE. After careful consideration, we feel that it has merit but does not fully meet PLOS ONE’s publication criteria as it currently stands. Therefore, we invite you to submit a revised version of the manuscript that addresses the points raised during the review process.

We look forward to receiving your revised manuscript.

Kind regards,

Krzysztof Durkalec-Michalski, Ph.D

Academic Editor

PLOS ONE

Reviewers' comments:

Reviewer's Responses to Questions

**Comments to the Author**

1. If the authors have adequately addressed your comments raised in a previous round of review and you feel that this manuscript is now acceptable for publication, you may indicate that here to bypass the “Comments to the Author” section, enter your conflict of interest statement in the “Confidential to Editor” section, and submit your "Accept" recommendation.

Reviewer #1: (No Response)

Reviewer #2: (No Response)

2. Is the manuscript technically sound, and do the data support the conclusions?

Reviewer #1: Yes

Reviewer #2: No

3. Has the statistical analysis been performed appropriately and rigorously? 

Reviewer #1: Yes

Reviewer #2: No

4. Have the authors made all data underlying the findings in their manuscript fully available?

Reviewer #1: No

Reviewer #2: Yes

5. Is the manuscript presented in an intelligible fashion and written in standard English?

Reviewer #1: Yes

Reviewer #2: No

6. Review Comments to the Author

Reviewer #1: The article is interesting. In my opinion, some aspects must be described more precisely.

1. The recommendations used in table 2 should be precisely described in methods, especially the estimation of calory needs.

2. Line 181: It is unclear how the authors chose the level of nutritional needs. The CHO differs from the cited recommendations and table 2, and it is unclear why they chose such levels of PROT and LIP. Also, the kcal level in table 1 is too low to satisfy the energy needs of athletes in this study.

1. In methods: the information about controlled and exchange diet should be precisely describe: precise characteristics and differences.

3. I propose not to use abbreviations in the titles of tables and sections.

4. Table 2. The authors should describe that the values presented in table 2 are calculated according to a different number of days: the weekdays –5 days, and pre-competition and competition days – one day. The way of calculation should be described in the methods and also in the information under the table.

5. Table 2. Under the table, the citations for the recommendations used should be included.

Reviewer #2: Minor revisions

1. Line 23: Add ‘players’ or ‘athletes’ after ‘elite female soccer’.

2. Line 33: Add comma before the word ‘while’.

3. Line 36: The expression ‘deficiency of haemoglobin’ seems to debatable. Consider using the expression ’participants presented decreased concentration of haemoglobin’ or ‘concentration of haemoglobin was below the reference level’.

4. Please explain all the abbreviations when they are first mentioned in the text. Keep using the same abbreviation for the corresponding expression throughout the whole manuscript (including tables, figures and text).

5. Line 53: The word ‘achieved’ seems to be out of the context and could be removed.

6. Line 55: Change ‘needs’ into ‘need’.

7. Line 80 – 83: No reference number is provided!

8. Line 83: The sentence ‘There are different nutritional intervention strategies to use with population’ is not clear.

9. Line 90: Use italics for ‘a priori’.

10. Unify using/not using spaces between special signs such as ‘+/=/±/’ within the whole manuscript.

11. Line 108: The ‘,’ is not necessary.

12. Line 198: Correct the writing of ‘guidelines’.

13. Using ‘LIP’ abbreviation for dietary fat is debatable. I would decline from using the abbreviation.

14. Table 2: Please change the language into English in the recommendations for fibre intake (now it states 22 - 25 g/día)

15. The recording of Reference list must to unified according to the guide lines of the journal.

Major revisions

1. The manuscript must undergo extensive English proof reading by native speaker.

2. The writing of units needs to be unified within the whole manuscript!

3. Line 59 – 60: Please consider referring to the following paper and recommendations: 10.1186/s12970-018-0242-y

4. Line 63 – 65: The given intake of PROT in females soccer players ‘1.7–2 g·kg-1 per day’ seems to be within the stated in the former part of manuscript (Line 61) recommendation range (i.e. 1.2 and 2.5 g·kg-1 per day), and not higher than recommendations (as Authors stated). The information requires verification and explanation.

5. Line 66 – 67: Please consider referring to the following paper and recommendations: 10.1186/s12970-018-0242-y

6. Two different abbreviations are used to ‘exchange diet’, namely ‘EG’ and ‘ED’ – it must be unified!

7. Figure 1 is not clear at all! It is not known which allocation refers to ‘CD’, and which to ‘ED/EG’. It must be improved.

8. Unify the number of decimal places for the same variables within the whole manuscript. The examples of mistakes with this respect can be found in lines 102 – 105, and other parts of the manuscript as well.

9. Please provide the questions that were included in the Diet Adherence Questionnaire (AQ). They are not provided at any part of the manuscript, and the questionnaire is new and not validated in the previous literature. Authors may provide the AQ as a supplementary material.

10. Line 119: Please provide the reference which describe the implemented methodology of dietary recording.

11. Line 124: Please provide the reference for EFSA guidelines that are mentioned in the text.

12. Line 130: There is a mistake in percentage classification of NK, to specify the mistake refers to the value ‘<25%’.

13. Provide the reference for ‘the International Society for the Advancement in Kinanthropometry’ (Line 140).

14. Line 161: How was euhydration state verified?

15. Consider moving the paragraph ‘Design and Procedures’ right after the paragraph ‘Participants’.

16. Provide the reference for the methodology of physical performance tests (i.e., running speed and CMJA).

17. Line 185: What does the abbreviation ‘CG’ stands for?

18. Statistical analysis – did all the variables actually present the normal distribution of the data? What was the procedure for non-normal distributed data?

19. The references for the recommendations provided in the Table 2 must be given.

20. Please correct the writing of the units in the Table 2.

21. Table 2: Please provide the exact equations that were used for estimating energy requirements!

22. Correct and unify decimal places (within the same variable) in the Table 2.

23. Table 3: The literature for each of reference range for each particular blood indices need to be provided.

7. PLOS authors have the option to publish the peer review history of their article (what does this mean?). If published, this will include your full peer review and any attached files.

Reviewer #1: No

Reviewer #2: No

---

## [Author Response · Author response to Decision Letter 1]

8 Mar 2023

Manuscript PONE-D-22-26202R1 (Review 1): answers to the referees.

Answers to reviewer 1

Please find our answers point by point. The changes are highlighted in the manuscript.

1. The recommendations used in table 2 should be precisely described in methods, especially the estimation of calory needs.

2. Line 181: It is unclear how the authors chose the level of nutritional needs. The CHO differs from the cited recommendations and table 2, and it is unclear why they chose such levels of PROT and LIP. Also, the kcal level in table 1 is too low to satisfy the energy needs of athletes in this study.

As you suggested, we have made some changes in the Methods section in order to make it clearer (Lines:130-138). Additionally, we have added some information in the table caption (table 2). 

1. In methods: the information about controlled and exchange diet should be precisely describe: precise characteristics and differences.

Following your advice, we have explained with more detail both dietetic interventions. (Lines: 142-154). (Methods section/Design and procedures)

2. I propose not to use abbreviations in the titles of tables and sections.

We have made some changes along the manuscript to solve this problem. 

3. Table 2. The authors should describe that the values presented in table 2 are calculated according to a different number of days: the weekdays –5 days, and pre-competition and competition days – one day. The way of calculation should be described in the methods and also in the information under the table.

In order to clarify this point, some changes have been made in the table (title and table caption) and in the text (Line: 159/ Methods section/Measures/ Dietary intake, nutritional knowledge and adherence questionnaires).

4. Table 2. Under the table, the citations for the recommendations used should be included.

The references have been included under the table.

Answers to reviewer 2

Please find our answers point by point

Reviewer #2: Minor revisions

1. Line 23: Add ‘players’ or ‘athletes’ after ‘elite female soccer’. 

2. Line 33: Add comma before the word ‘while’.

5.Line 53: The word ‘achieved’ seems to be out of the context and could be removed. 

6. Line 55: Change ‘needs’ into ‘need’. 

9. Line 90: Use italics for ‘a priori’. 

10. Unify using/not using spaces between special signs such as ‘+/=/±/’ within the whole manuscript. 

11. Line 108: The ‘,’ is not necessary. 

12. Line 198: Correct the writing of ‘guidelines’. 

14. Table 2: Please change the language into English in the recommendations for fibre intake (now it states 22 - 25 g/día)

We have followed all the instructions given for the reviewer. Some of them were trivial mistakes that have been corrected. Thank you very much.

3.Line 36: The expression ‘deficiency of haemoglobin’ seems to debatable. Consider using the expression ’participants presented decreased concentration of haemoglobin’ or ‘concentration of haemoglobin was below the reference level’.

Thank you very much for your helpful advice. It sounds much better that way.

4.Please explain all the abbreviations when they are first mentioned in the text. Keep using the same abbreviation for the corresponding expression throughout the whole manuscript (including tables, figures and text). 

The entire manuscript has been reviewed to correct any mistakes.

7. Line 80 – 83: No reference number is provided! 

The reference number is now provided. (Line: 81)

8. Line 83: The sentence ‘There are different nutritional intervention strategies to use with population’ is not clear. 

The sentence has been rewritten to make it clearer. (Line: 84).

13. Using ‘LIP’ abbreviation for dietary fat is debatable. I would decline from using the abbreviation. 

Following your suggestion, we have replaced all the “LIP” abbreviation for the word “fat” which sounds better.

15. The recording of Reference list must to unified according to the guide lines of the journal. 

The reference list has been unified.

Reviewer #2: Major revisions 

1.The manuscript must undergo extensive English proof reading by native speaker. 

The manuscript has been reviewed, and many changes have been made to improve the language. However, not all the changes have been highlighted at this point to make the text easier to read.

2.The writing of units needs to be unified within the whole manuscript! 

6.Two different abbreviations are used to ‘exchange diet’, namely ‘EG’ and ‘ED’ – it must be unified! 

8. Unify the number of decimal places for the same variables within the whole manuscript. The examples of mistakes with this respect can be found in lines 102 – 105, and other parts of the manuscript as well.

The entire manuscript has been reviewed to correct any mistakes.

3.Line 59 – 60: Please consider referring to the following paper and recommendations: 10.1186/s12970-018-0242-y 

5. Line 66 – 67: Please consider referring to the following paper and recommendations: 10.1186/s12970-018-0242-y 

Thank you very much for your advice. We have added the reference mentioned.

4. Line 63 – 65: The given intake of PROT in females soccer players ‘1.7–2 g·kg-1 per day’ seems to be within the stated in the former part of manuscript (Line 61) recommendation range (i.e. 1.2 and 2.5 g·kg-1 per day), and not higher than recommendations (as Authors stated). The information requires verification and explanation. 

Thank you very much for the appreciation. There was an error in the text that has been corrected. 

7. Figure 1 is not clear at all! It is not known which allocation refers to ‘CD’, and which to ‘ED/EG’. It must be improved. 

Thank you very much for your assessment. We have added the corresponding group allocation in the participan flow diagram to improve it. 

9. Please provide the questions that were included in the Diet Adherence Questionnaire (AQ). They are not provided at any part of the manuscript, and the questionnaire is new and not validated in the previous literature. Authors may provide the AQ as a supplementary material. 

The document has been attached as a supplementary file.

10. Line 119: Please provide the reference which describe the implemented methodology of dietary recording.

11. Line 124: Please provide the reference for EFSA guidelines that are mentioned in the text. 

13. Provide the reference for ‘the International Society for the Advancement in Kinanthropometry’ (Line 140).

16. Provide the reference for the methodology of physical performance tests (i.e., running speed and CMJA).

These references have been included. (Reference list, numbers 19, 21, 24, 25 and 26, respectively).

12. Line 130: There is a mistake in percentage classification of NK, to specify the mistake refers to the value 

Sorry, it was a trivial mistake that has been corrected.

14. Line 161: How was euhydration state verified? 

The euhydration status of the players prior to the blood analyses was not specifically verified. However, those players strictly followed the protocols established by the laboratory resposinble of the test, which teoretically guarantee proper hydration status. In order to avoid misunderstunding, the word “euhydrated” has been omitted. (Line 202)

15. Consider moving the paragraph ‘Design and Procedures’ right after the paragraph ‘Participants’. 

The change suggested has been made.

17. Line 185: What does the abbreviation ‘CG’ stands for?

 All the abreviations have been revised in order to avoid misunderstandings.

18. Statistical analysis – did all the variables actually present the normal distribution of the data? What was the procedure for non-normal distributed data? 

All the variables presented a normal distribution of the data, therefore, we did not use any procedure for non normal distributed data. We have made some changes in the text in order to clarify this point (Line 211).

19. The references for the recommendations provided in the Table 2 must be given. 

The references have been included under the table.

20. Please correct the writing of the units in the Table 2. 

The entire manuscript has been reviewed to correct the units.

21. Table 2: Please provide the exact equations that were used for estimating energy requirements! 

We have made some changes in the Methods section in order to make it clearer (Lines:130-138). Additionally, we have added some information in the table caption.

22. Correct and unify decimal places (within the same variable) in the Table 2. 

The entire manuscript has been reviewed to unify decimal places.

23. Table 3: The literature for each of reference range for each particular blood indices need to be provided. 

The reference ranges used for the parameters of the blood análisis are established by the laboratory responsable for peforming them. That is the reason why we have not included any reference to them, as we understand that they are sufficiently validated intervals.

---

## [Decision Letter · Decision Letter 2]

13 Apr 2023

PONE-D-22-26202R2Effects of exchange vs. controlled diet on biochemical, body composition and functional parameters in elite female soccer playersPLOS ONE

Dear Dr. Antón San Atanasio,

Thank you for submitting your manuscript to PLOS ONE. After careful consideration, we feel that it has merit but does not fully meet PLOS ONE’s publication criteria as it currently stands. Therefore, we invite you to submit a revised version of the manuscript that addresses the points raised during the review process.

We look forward to receiving your revised manuscript.

Kind regards,

Krzysztof Durkalec-Michalski, Ph.D

Academic Editor

PLOS ONE

Reviewers' comments:

Reviewer's Responses to Questions

**Comments to the Author**

1. If the authors have adequately addressed your comments raised in a previous round of review and you feel that this manuscript is now acceptable for publication, you may indicate that here to bypass the “Comments to the Author” section, enter your conflict of interest statement in the “Confidential to Editor” section, and submit your "Accept" recommendation.

Reviewer #1: All comments have been addressed

Reviewer #2: (No Response)

2. Is the manuscript technically sound, and do the data support the conclusions?

Reviewer #1: (No Response)

Reviewer #2: Partly

3. Has the statistical analysis been performed appropriately and rigorously? 

Reviewer #1: Yes

Reviewer #2: No

4. Have the authors made all data underlying the findings in their manuscript fully available?

Reviewer #1: Yes

Reviewer #2: No

5. Is the manuscript presented in an intelligible fashion and written in standard English?

Reviewer #1: Yes

Reviewer #2: Yes

6. Review Comments to the Author

Reviewer #1: (No Response)

Reviewer #2: Minor revisions

1. Please unify writing of heamoglobin/hemoglobin within the whole manuscript (an example of various writing can be found in the lines 36 – 37; please verify the writing of the related words, e.g. hematopoietic).

2. Line 39: I suggest replacing “parameters” with “indices”. Same in the line 122.

3. Do not use the space between the value and “%” sign.

4. Line 54: I suggest adding “physical” before “performance”.

5. Line 60: There is no position no. 8 in the reference list.

6. I suggest using 5 – 7 gCHO·kgBM-1∙day-1 instead of 5 – 7 g·kg-1 per day (I suggest analogous changes within the whole manuscript with regard to “similar” units).

7. Line 64: You can use “PROT” for protein (please revise the whole manuscript with this respect)

8. Line 67-68: I can’t agree with the statement.

9. Line 115: Figure 1 shows the flow of participants in the study – the sentence is redundant.

10. Line 116. Correct the writing of the title of the Figure 1.

11. Line 130: Correct the writing of “kg”.

12. Authors need to indicate in the text of the manuscript that the “Adherence questionnaire” is provided as an attachment. It should be provided also in English language version.

13. Table 1. Replace “,” with “.”.

14. Lines 232-237: I suggest providing the full names of mineral (at least when first mention in the text).

15. The reference list is still in mess. It must be improved.

16. Line 387: “dietetic” must be replace with “dietary”.

17. I recommend providing results of vertical jump in cm.

Major revisions

1. Line 105: How was the randomization process performed?

2. Line 124: Why was dietary intake recorded only PRE? It seems to be a serious methodological concern.

3. Line 130 – 131: Energy intake during dietary intervention was calculated based of an average body mass. It is a serious impropriety. It must have been adjusted to each participant individual body mass. Same should be done with regard to CHO, PROT and FAT.

4. Lines 137:138. I do not agree with the statement “A sudden increase in CHO intake would have been difficult to adhere to”. The assumptions of implemented dietary intervention should take into account actual needs for CHO and the intake of CHO during intervention should be in line with current scientific-based recommendations. This fact must be discussed in the Discussion paragraph as a serious limitation of the study.

5. Line 205-206: “The evaluation was based on the reference range (table 3) indicated by the 206 biochemical analysis laboratory” – still, the literature references for the ranges must be provided.

6. Statistical analysis: I would kindly ask the Authors to provide the raw data of study results for the for inspection of reviewers.

7. Table 2: It is not necessary to provide two decimal places with regard to the intake of CHO, PROT, fat, fatty acids when expressed as g∙day-1; fiber, calcium, phosphorus, magnesium, iron, vitamin C.

8. Lines 300-302: “Regarding biochemical values, a well-established controlled diet, including Fe and vit C consumption (34), has shown to increase haemoglobin concentration but no other parameters” - there is no justification for such a statement, while the intakes of Fe and Vit C were not monitored, neither during the intervention, nor POST-intervention. We don not know the exact reason for increase in haemoglobin concentration.

9. Lines 310-325: Author underline the importance of CHO in the nutrition of female soccer players and the assumptions of the implemented dietary interventions (EG and CD) did not ensure provision of CHO according to recommendations.

10. Sample size calculation: Authors stated “ The estimated sample size was 17 participants (actual power = 0.802), but considering possible dropouts, we enrolled 19 participants in this study” (lines 226-227). While in the Figure 1 it is said that 21 participants were enrolled, 20 were randomized, and there was 1 drop-out. It is thus inconsistent.

7. PLOS authors have the option to publish the peer review history of their article (what does this mean?). If published, this will include your full peer review and any attached files.

Reviewer #1: No

Reviewer #2: No

---

## [Author Response · Author response to Decision Letter 2]

22 May 2023

Manuscript PONE-D-22-26202R2(Revision 2): answers to the referees.

Answers to reviewer 2

Please find our answers point by point. The changes are highlighted in the manuscript.

MINOR REVISIONS

1. Please unify writing of heamoglobin/hemoglobin within the whole manuscript (an example of various writing can be found in the lines 36 – 37; please verify the writing of the related words, e.g. hematopoietic). 

2. Line 39: I suggest replacing “parameters” with “indices”. Same in the line 122. 

3. Do not use the space between the value and “%” sign. 

4. Line 54: I suggest adding “physical” before “performance”. 

We have followed all the instructions provided by the reviewer. Some of them were trivial mistakes that have been corrected. Thank you very much for your guidance.

5. Line 60: There is no position no. 8 in the reference list. 

The reference list was reviewed, and a new reference (no.8) was added, as it was unintentionally skipped. 

6. I suggest using 5 – 7 gCHO·kgBM-1∙day-1 instead of 5 – 7 g·kg-1 per day (I suggest analogous changes within the whole manuscript with regard to “similar” units). 

7. Line 64: You can use “PROT” for protein (please revise the whole manuscript with this respect) 

The entire manuscript has been reviewed and corrections have been made to address your suggestions. 

8. Line 67-68: I can’t agree with the statement. 

In the statement made in lines 67-68, we simply refer to what has been indicated in previous studies, which have observed that the fat intake in female soccer players falls within the recommended range of 20-35%. However, within that intake, the consumption of saturated fats generally exceeds the recommended limit of 10%. This is also observed in the present study, although it is not mentioned explicty yet, as it will be included in the Results and Discussion sections later on.

9. Line 115: Figure 1 shows the flow of participants in the study – the sentence is redundant. 

10. Line 116. Correct the writing of the title of the Figure 1. 

Following your recomendations, the sentence has been omitted and the title corrected.

11. Line 130: Correct the writing of “kg”. 

13.Table 1. Replace “,” with “.”. 

16. Line 387: “dietetic” must be replace with “dietary”. 

All of these minor corrections have already been made. 

12. Authors need to indicate in the text of the manuscript that the “Adherence questionnaire” is provided as an attachment. It should be provided also in English language version. 

As you suggested, it is indicated in lines 176-177 and the adherence questionnaire is now provided in English. Previously, it was provided in both Spanish and English.

14. Lines 232-237: I suggest providing the full names of mineral (at least when first mention in the text). 

The entire manuscript has been reviewed and corrections have been made to follow your suggestion. 

15. The reference list is still in mess. It must be improved. 

The entire reference list has been thoroughly reviewed and several mistakes have been corrected.

17. I recommend providing results of vertical jump in cm. 

As you suggested, the results are now provided in cm (Figure 2).

MAJOR REVISIONS

1. Line 105: How was the randomization process performed? 

As it is an important question, we have added an explanation in the text (lines 107-108). A simple randomization method was employed by generating a table of random numbers without repetition. Participants with even numbers were assigned to the CG, while participants with odd numbers were assigned to the EG.

2. Line 124: Why was dietary intake recorded only PRE? It seems to be a serious methodological concern. 

While it could be considered a serious methodological concern, the main objective of the study was not to evaluate changes in dietary intake post-intervention. Instead, the focus was on assessing the nutritional energy intake pre-intervention for comparison with general recommendations and for the purpose of implementing an appropriate nutritional intervention. However, it is important to acknowledge that not evaluating post-intervention dietary intake may limit the assessment of the intervention’s effectiveness in terms of dietary changes.

3. Line 130 – 131: Energy intake during dietary intervention was calculated based of an average body mass. It is a serious impropriety. It must have been adjusted to each participant individual body mass. Same should be done with regard to CHO, PROT and FAT. 

It is important to address the concern raised by the reviewer regarding individual variations in body mass. However, it is crucial to consider the context of the present study, which was an initial group intervention aimed at assessing suitable intervention methodologies based on adherence and nutritional knowledge questionnaire results. Given that most players had weights close to the average, using the average weight to meet the nutritional requirements of all players was initially considered appropriate. Nevertheless, upon reflection, we acknowledge that adjusting energy intake to each participant’s individual body mass would have provided a more precise and tailored approach to meet their specific needs. We apologize for the oversight and recognize the importance of individualizing nutritional interventions in future studies. In light of the results obtained in the present study, we intend to implement interventions based on each player’s individual objectives, periodized according to the training plan and the stage of the season. Additionally, we plan to incorporate periodization based on each player’s menstrual cycle.

Thank you for bringing this concern to our attention, and we appreciate the opportunity to improve our methodology for future research.

4. Lines 137:138. I do not agree with the statement “A sudden increase in CHO intake would have been difficult to adhere to”. The assumptions of implemented dietary intervention should take into account actual needs for CHO and the intake of CHO during intervention should be in line with current scientific-based recommendations. This fact must be discussed in the Discussion paragraph as a serious limitation of the study. 

Thank you for your feedback. As you suggested, we have included a paragraph (lines 397-405) in the limitations section of the study (Discussion) to address the issue of carbohydrate intake during the intervention, which was slightly lower than the current minimum recommendation. This decision was made to ensure gradual changes in carbohydrate intake during the intervention, taking into consideration the high prevalence of eating disorders and the focus on weight control among female athletes. By avoiding a sudden increase in intake, we aimed to create a more manageable and sustainable approach for the participants. By acknowledging this as a limitation, we emphasize the importance of future research considering the participant’s actual needs, aligning their carbohydrate intake with current scientific-based recommendations during intervention.

5. Line 205-206: “The evaluation was based on the reference range (table 3) indicated by the 206 biochemical analysis laboratory” – still, the literature references for the ranges must be provided. 

Thank you for your additional comments regarding the reference ranges used in the study. The laboratory used for the analisis was Synlab, a clinical laboratory accredited by national accreditation bodies such as DAkkS in Germany, SAS in Switzerland, and ENAC in Spain. These bodies evaluate and accredit laboratories to ensure their technical competence and compliance with relevant international and national standards. You can find more information about Synlab and its accreditations on their oficial website: https://synlab.es . Unfortunately, specific literature references for the reference ranges used in the clinical laboratory were not included, something that is commeon in scientific articles of this kind, as the information provided is based on the laboratory’s internal standards and protocols in accordance with its accreditation.

6. Statistical analysis: I would kindly ask the Authors to provide the raw data of study results for the for inspection of reviewers. 

In order to facilitate the comprehension of Figure 2, the raw data have been attached as a supplementary file, entitled FIGURE 2 DATABASE.

7. Table 2: It is not necessary to provide two decimal places with regard to the intake of CHO, PROT, fat, fatty acids when expressed as g∙day-1; fiber, calcium, phosphorus, magnesium, iron, vitamin C. 

Following your suggestion, we have reviewed the entire table 2 and made the necessary changes. 

8. Lines 300-302: “Regarding biochemical values, a well-established controlled diet, including Fe and vit C consumption (34), has shown to increase haemoglobin concentration but no other parameters” - there is no justification for such a statement, while the intakes of Fe and Vit C were not monitored, neither during the intervention, nor POST-intervention. We don not know the exact reason for increase in haemoglobin concentration.

This statement has been omitted in this section, as it should exclusively focus on presenting results without interpretations. However, we have added a possible explanation in the discussion section (lines 372-378) to address this aspect. 

9. Lines 310-325: Author underline the importance of CHO in the nutrition of female soccer players and the assumptions of the implemented dietary interventions (EG and CD) did not ensure provision of CHO according to recommendations. 

As stated previously in point 4 of the major revisions, we have acknowledged this as a limitaion os the study. A paragraph has been added in the duscussion section to adress this subject.

10. Sample size calculation: Authors stated “ The estimated sample size was 17 participants (actual power = 0.802), but considering possible dropouts, we enrolled 19 participants in this study” (lines 226-227). While in the Figure 1 it is said that 21 participants were enrolled, 20 were randomized, and there was 1 drop-out. It is thus inconsistent. 

Following your comment, we have carefully reviewed both the text and Figure 1 and identified an error in the graphic. We have made the necessary changes to correct it. Additionally, we have made clarifications in the text (lines 220-234) to provide a clearer explanation of the sample size calculation.

---

## [Decision Letter · Decision Letter 3]

6 Jun 2023

PONE-D-22-26202R3Effects of exchange vs. controlled diet on biochemical, body composition and functional parameters in elite female soccer playersPLOS ONE

Dear Dr. San Atanasio,

Thank you for submitting your manuscript to PLOS ONE.After careful consideration, we feel that it has merit but does not fully meet PLOS ONE’s publication criteria as it currently stands. Your submission has been largely corrected, but the reviewer points out that all the data necessary for a full evaluation of the work are still not included. Please make additions in this regard. Therefore, we invite you to submit a revised version of the manuscript that addresses the points raised during the review process.

We look forward to receiving your revised manuscript.

Kind regards,

Krzysztof Durkalec-Michalski, Ph.D

Academic Editor

PLOS ONE

Reviewers' comments:

Reviewer's Responses to Questions

**Comments to the Author**

1. If the authors have adequately addressed your comments raised in a previous round of review and you feel that this manuscript is now acceptable for publication, you may indicate that here to bypass the “Comments to the Author” section, enter your conflict of interest statement in the “Confidential to Editor” section, and submit your "Accept" recommendation.

Reviewer #2: (No Response)

2. Is the manuscript technically sound, and do the data support the conclusions?

Reviewer #2: Yes

3. Has the statistical analysis been performed appropriately and rigorously? 

Reviewer #2: I Don't Know

4. Have the authors made all data underlying the findings in their manuscript fully available?

Reviewer #2: No

5. Is the manuscript presented in an intelligible fashion and written in standard English?

Reviewer #2: Yes

6. Review Comments to the Author

Reviewer #2: **Although Authors addressed carefully most of my previous revisions, one of the main major revision is still not resolved. Namely, Authors provided the data set solely to support Figure 2. Remaining data sets are not provided. Thus, I would kindly ask to provide data supporting all the remaining tables and figures for review process only.**

7. PLOS authors have the option to publish the peer review history of their article (what does this mean?). If published, this will include your full peer review and any attached files.

Reviewer #2: No

---

## [Author Response · Author response to Decision Letter 3]

25 Jun 2023

-Manuscript PONE-D-22-26202: Effects of exchange vs. controlled diet on biochemical, body composition and functional parameters in elite female soccer players (Revision 3): answers to the referees.

Answer to reviewer 2

COMMENTS

Although Authors addressed carefully most of my previous revisions, one of the main major revision is still not resolved. Namely, Authors provided the data set solely to support Figure 2. Remaining data sets are not provided. Thus, I would kindly ask to provide data supporting all the remaining tables and figures for review process only.

We sincerely apologize for our previous response, because we misunderstood your requirements. We thought you were only referring to raw data for Figure 2. As per your suggestion, we have now attached a supplementary file titled “GLOBAL DATABASE”.

---

## [Decision Letter · Decision Letter 4]

12 Jul 2023

Effects of exchange vs. controlled diet on biochemical, body composition and functional parameters in elite female soccer players

PONE-D-22-26202R4

Dear Prof. Anton,

We’re pleased to inform you that your manuscript has been judged scientifically suitable for publication and will be formally accepted for publication once it meets all outstanding technical requirements.

Kind regards,

Krzysztof Durkalec-Michalski, Ph.D

Academic Editor

PLOS ONE

Additional Editor Comments (optional):

The work should standardize the notation of units (e.g. superscripts of "-1")

Reviewers' comments:

Reviewer's Responses to Questions

**Comments to the Author**

1. If the authors have adequately addressed your comments raised in a previous round of review and you feel that this manuscript is now acceptable for publication, you may indicate that here to bypass the “Comments to the Author” section, enter your conflict of interest statement in the “Confidential to Editor” section, and submit your "Accept" recommendation.

Reviewer #2: All comments have been addressed

2. Is the manuscript technically sound, and do the data support the conclusions?

Reviewer #2: Yes

3. Has the statistical analysis been performed appropriately and rigorously? 

Reviewer #2: Yes

4. Have the authors made all data underlying the findings in their manuscript fully available?

Reviewer #2: Yes

5. Is the manuscript presented in an intelligible fashion and written in standard English?

Reviewer #2: Yes

6. Review Comments to the Author

Reviewer #2: (No Response)

7. PLOS authors have the option to publish the peer review history of their article (what does this mean?). If published, this will include your full peer review and any attached files.

Reviewer #2: No

---

## [Editor Report · Acceptance letter]

14 Jul 2023

PONE-D-22-26202R4 

Effects of exchange vs. controlled diet on biochemical, body composition and functional parameters in elite female soccer players 

Dear Dr. San Atanasio:

I'm pleased to inform you that your manuscript has been deemed suitable for publication in PLOS ONE. Congratulations! Your manuscript is now with our production department. 

Kind regards, 

on behalf of

Dr. Krzysztof Durkalec-Michalski 

Academic Editor

PLOS ONE